# Restoring confidence in return to work: A qualitative study of the experiences of persons with exhaustion disorder after a dialogue-based workplace intervention

Maria Strömbäck[1,2]*, Anncristine Fjellman-Wiklund[1], Sara Keisu[1], Marine Sturesson[3], Therese Eskilsson[1,4]

1 Department of Community Medicine and Rehabilitation, Physiotherapy, Umeå University, Umeå, Sweden, 2 Department of Clinical Science, Psychiatry, Umeå University, Umeå, Sweden, 3 Department of Community Medicine and Rehabilitation, Occupational therapy, Umeå University, Umeå, Sweden, 4 Department of Public Health and Clinical Medicine, Section of Sustainable Health, Umeå University, Umeå, Sweden

* maria.stromback@umu.se

**Data Availability Statement:** Data cannot be shared publicly because of the Swedish Data

## Abstract

### Background

Stress-induced exhaustion disorder (SED) is a primary cause for sickness absence among persons with mental health disorders in Sweden. Interventions involving the workplace, and supporting communication between the employee and the supervisor, are proposed to facilitate return to work (RTW). The aim of this study was to explore experiences of persons with SED who participated in a dialogue-based workplace intervention with a convergence dialogue meeting performed by a rehabilitation coordinator.

### Methods

A qualitative design based on group interviews with 15 persons with SED who participated in a 24-week multimodal rehabilitation program was used. The interviews were analyzed with the methodology of grounded theory.

### Results

The analysis resulted in a theoretical model where the core category, *restoring confidence on common ground*, represented a health promoting process that included three phases: *emotional entrance*, *supportive guidance*, and *empowering change*. The health promoting process was represented in participant experiences of personal progress and safety in RTW.

### Conclusions

The intervention built on a health-promoting pedagogy, supported by continuous guidance from a rehabilitation coordinator and structured convergence dialogue meetings that enhanced common communication and collaboration with the supervisor and others

Protection Act (1998:204), which does not permit sharing of de-identified and sensitive data on humans (like in our interviews). Data are available from Umeå University, Sweden (contact via medfak@umu.se) for researchers who meet the criteria for access to confidential data.

**Funding:** This work received support from AFA Insurance, and Region Västerbotten in Sweden. TE received the award with grant number: AFA Insurance Dnr 150274. The funders had no role in study design, data collection and analysis, decision to publish, or preparation of the manuscript. AFA Insurance: https://www.afaforsakring.se/andra-sprak/engelska/ The Västerbotten County Council: https://regionvasterbotten.se/informationsmaterial-och-riktlinjer/andra-sprak/engelska-eller-english The funders had no role in study design, data collection and analysis, decision to publish, or preparation of the manuscript.

involved in the RTW process. The intervention balanced relationships, transferred knowledge, and changed attitudes about SED among supervisors and colleagues in the workplace. The inclusion of a rehabilitation coordinator in the intervention was beneficial by enhancing RTW and bridging the gaps between healthcare, the workplace, and other organizational structures. In addition, the intervention contributed to a positive re-orientation towards successful RTW instead of an endpoint of employment. In a prolonged process, a dialogue-based workplace intervention with convergence dialogue meetings and a rehabilitation coordinator may support sustainable RTW for persons with SED.

## Introduction

Work-related stress and burnout are common throughout Europe [1, 2], have an increased risk for long-term sick leave [3], and increasing costs for society [2]. Burnout may lead to impaired quality of life for the individual through withdrawal, isolation and discrimination [4, 5].

Definition of burnout varies between countries [2]. In Sweden, stress-induced exhaustion disorder or exhaustion disorder (SED) (SED; F43.8A, ICD-10-SE) is classified as an illness and the diagnosis is used in Swedish healthcare practice [6]. SED is characterized by pronounced physical and mental exhaustion as a consequence of at least six months of identifiable stressors [7]. Mental symptoms such as burnout, anxiety and depression [8], and somatic symptoms such as nausea and headaches [9] are reported frequently in patients with SED. The prevalence of mental and somatic symptoms is high irrespective of gender, age or education [8, 9]. Cognitive impairments such as problems with memory and executive functions are other commonly reported symptoms [10].

Multiple stressors, including work and non-work stressors, are considered important in the onset of SED [8, 11]. The most frequently reported stressors are quantitative and emotional demands at work and private relational conflicts, with similar patterns for women and men [11]. Prominent work-related psychosocial risk factors affecting the onset of burnout are high job demands, low job control, tense social relationships at work, injustice at work, and low rewards [2, 12, 13]. Thus, work-related risk factors are associated with an increased risk for emotional exhaustion, while high levels of job support and workplace justice are protective and reduce the risk for emotional exhaustion [13]. A meta-synthesis of qualitative research investigating return to work (RTW) among persons with common mental disorders found that obstacles and facilitators were related to handling personal demands, the need for social support at the workplace, and to bridge contacts between the social and rehabilitation systems. Additional factors found to affect RTW were difficulties in deciding the appropriate time to resume work and when to implement adjustments for RTW [14].

Development of interventions that enhance RTW for persons with burnout and SED is essential. Work-directed interventions for burnout and mental disorders have most promising results for reduced sickness absence, even though the results are inconclusive [15–18]. Improved health symptoms do not automatically lead to improved RTW, and this confirms the need to develop interventions that bridge rehabilitation with organizational arrangements [13, 15, 18] and prevent the onset of SED at workplaces [6]. Interventions that initiate worker-supervisor communication show interesting and positive outcomes [16, 18, 19, 20]. High feasibility has been shown for a participatory workplace intervention for persons with distress that involves the supervisor in a three-step communication process performed by an occupational health service RTW coordinator [21]. That intervention showed no lasting effect on RTW but appeared to be effective for employees who intended to RTW despite symptoms [22]. In

Sweden, Karlson et al [23] developed a workplace-oriented intervention for persons on sick leave due to burnout because of work-related stress. The intervention is a structured three-step interview model including a convergence dialogue meeting. The aim is to initiate a dialogue between the employee and the supervisor in order to find concrete solutions that facilitate RTW. The structured three-step interview model shows beneficial effects in RTW [23] and over time for participants who were 45 years or younger [24]. In another Swedish study, a workplace dialogue intervention (WDI) that included a convergence dialogue meeting was compared to acceptance and commitment therapy (ACT), a combination of ACT and WDI, and treatment as usual for persons on sick leave because of depression, anxiety disorders or SED [25]. None of the interventions were significantly more effective than another to reduce days of sick leave. However, when different diagnostic groups were compared, persons diagnosed with SED who received WDI had fewer sick days compared to treatment as usual [25]. A study evaluating the effect of including a convergence dialogue meeting as part of cognitive behavioral therapy (CBT) to facilitate RTW for persons with common mental disorders concluded that the convergence dialogue meeting did not shorten days of sick leave before full RTW [26]. However, studies supporting improved worker-supervisor communication demonstrate the need to tailor interventions to match diagnosis-specific needs [25–27], and the need for specific competencies to improve communication between healthcare, the workplace, and social systems involved in the RTW process [14, 21, 25].

In Sweden, a new competency in healthcare has been introduced. A rehabilitation coordinator has the role of supporting patients on sick leave during their rehabilitation and coordinating contacts with other involved stakeholders in the RTW process [28]. A rehabilitation coordinator is available in every Swedish region, in primary care and specialist healthcare. The rehabilitation coordinator has professional background and working experience in healthcare (medicine, psychology, social work, physiotherapy or occupational therapy). The rehabilitation coordinator's job is to have close collaboration with medical doctors and other team members involved in the patient's treatment and rehabilitation plan, including mapping of medical needs, current work and earlier periods of sick leave. Depending on setting, the rehabilitation coordinator has a broad range of knowledge in the natural history of different diseases and conditions (for example SED), the rehabilitation process, assessments made in healthcare and insurance medicine, the labor market, and work-related rehabilitation efforts [28]. Evaluations in primary healthcare by the rehabilitation coordinator show opportunities to decrease days of sickness absence, a faster RTW process, and improved quality of life compared to a control group [29]. However, structured guidelines are needed on which methods the rehabilitation coordinator should use.

In this study, we focused on investigating experiences of a dialogue-based workplace intervention with convergence dialogue meetings that was performed by a rehabilitation coordinator. The intervention was influenced by evaluation of a previous workplace-oriented intervention by Karlson et al [23, 24], and further developed to include a structure for planning, follow-ups and extended support for RTW. The rehabilitation coordinator, who organized and coordinated internal and external collaborations in the RTW process, performed the intervention. The aim of this study was to explore experiences of persons with SED who participated in a dialogue-based workplace intervention with a convergence dialogue meeting performed by a rehabilitation coordinator.

## Methods

### Study design

A qualitative research design with semi-structured group interviews was used. The group interviews were analyzed with a modified grounded theory, illustrating social processes and

generating the development of a theoretical model. Grounded theory is suitable when a new intervention is evaluated and complex conditions about health and illness are studied [30, 31], such as in our study including persons with SED. Grounded theory is theoretically based in social constructivism, and the grounded theory method used in this study is inspired by the methodology of Charmaz and symbolic interactionism [30]. Symbolic interactionism is a dynamic theoretical perspective that assumes that people construct identities, social rules and reality through interaction [30].

## Ethical statement

Ethical approval was confirmed by the Regional Ethical Review Board in Umeå, Sweden (2015/49-31Ö). All participants received verbal and written information about the study before deciding on participation. Participants were informed that their participation was voluntary and that they could withdraw at any time without declaring a reason. Participants were assured confidentiality and anonymity in presentation of the results. After participants agreed to participate, they signed a written consent form.

## Study context and participants

The study was conducted at the Stress Rehabilitation Clinic at the University Hospital of Umeå, Sweden. The clinic specializes in rehabilitation of persons with SED. The treatment was based on a 24-week multimodal rehabilitation program, including a dialogue-based workplace intervention with convergence dialogue meetings intended to promote RTW. Participants were recruited from the multimodal rehabilitation program.

The multimodal rehabilitation program included group-based cognitive behavioral therapy (CBT) to support behavioral changes and to develop strategies to promote health. The CBT groups consisted of eight persons in each group who met weekly for 22 three-hour sessions. Each person had two individual meetings with the group leader to set and evaluate individual goals. Examples of individual goals could include development of strategies to improve one's capability to handle stressful situations, manage high workloads, set limits or take regular breaks. The multimodal rehabilitation program was previously evaluated in randomized controlled trials [32, 33].

Between March and October 2016, 24 persons from seven CBT groups in the multimodal rehabilitation program were asked to participate in group interviews [34]. The persons included in this study each had a confirmed diagnosis of SED [6], were between the working ages of 18 to 60 years, had current employment, and were on at least 50% sick leave.

Fifteen persons (13 women and two men) agreed to be interviewed; nine declined. The seven group interviews were composed from within the CBT groups, which meant that study participants in each group interview knew each other. One group consisted of three participants and the remaining six groups consisted of two participants each. The participants were between 33 and 56 years of age, nine worked with people, three worked with things, and three worked with data.

## The dialogue-based workplace intervention with a convergence dialogue meeting

The dialogue-based workplace intervention with a convergence dialogue meeting was a structured three-step interview model with follow-ups, and was performed and organized by a rehabilitation coordinator at the Stress Rehabilitation Clinic. The intervention aimed to facilitate dialogue between the participant and supervisor responsible for rehabilitation at the workplace, and other involved stakeholders in the RTW process. The rehabilitation coordinator's

role was to be a liaison between involved persons in the intervention. The rehabilitation coordinator had specific knowledge about SED and social insurance medicine, and was educated and trained in implementing convergence dialogue meetings through a written manual and practical guidance from experienced personnel in the research group and with a background in occupational health.

The first step in the structured three-step interview model was an individual interview of the participant by the rehabilitation coordinator. At that point, the participant gave informed consent to contact the supervisor. In the second step, the supervisor was interviewed at the Stress Rehabilitation Clinic, by phone, or by video link. The structured interviews included questions about expectations and concerns for rehabilitation, perceived main causes for the participant´s sick leave, possible ways to adjust the work situation, suggestions on how to facilitate RTW, and motivation and confidence for RTW. In addition, the supervisor answered questions about systematic work environment issues, access to occupational healthcare and if specific actions were planned. In the third step, the rehabilitation coordinator performed a convergence dialogue meeting with the participant and supervisor. The participant could invite a representative from the workplace or trade union. The convergence dialogue meeting lasted for about 1.5 hours and the focus was to frame constructive solutions to facilitate and prepare for a sustainable RTW process. The interview guide to the structured three-step interview model is attached as (S1 and S2 Appendixes). The convergence dialogue meeting built upon a health promotion approach (Fig 1).

This approach focused on the work environment, including physical, organizational and social factors, when discussing work tasks in relation to work ability. Participant and supervisor perspectives were taken into consideration in adjusting work tasks that might help facilitate RTW. Adjustments were made primarily on the basis of common problems in SED, such as reduced cognitive ability and the need to balance activity and recovery at work.

The convergence dialogue meeting resulted in a written plan that included goals, actions, follow-ups, and division of responsibility between the participant and supervisor. The supervisor was responsible for initiating continuous follow-ups. The written plan functioned as an agreement between involved parties and was revised if duties or work hours changed, or if the goals needed to be adjusted based on the participant's abilities. A copy of the written plan was distributed to the participant, the supervisor, and the Social Insurance Office. The rehabilitation coordinator made a follow-up at least once and then as needed during the multimodal rehabilitation program.

## Group interviews

Group interviews were done after the dialogue-based workplace intervention with convergence dialogue meetings and the multimodal rehabilitation program were finalized. Organization of the group interviews was inspired by focus group methodology [34]. A psychologist and an assistant (occupational therapist or physician) performed the group interviews. None of the interviewers was involved in the participants' rehabilitation. The group interviews followed a semi-structured guide, including questions related to starting the return to work process, design of the intervention, convergence dialogue meeting, role of the rehabilitation coordinator, feasibility of the plan, and future expectations. Each group interview lasted about 1–1.5 hour, was audio-recorded, and transcribed verbatim.

## Data analysis

Data from the transcribed group interviews were analyzed using a modified grounded theory method with a focus on constant comparisons [30, 31]. The modification entailed all

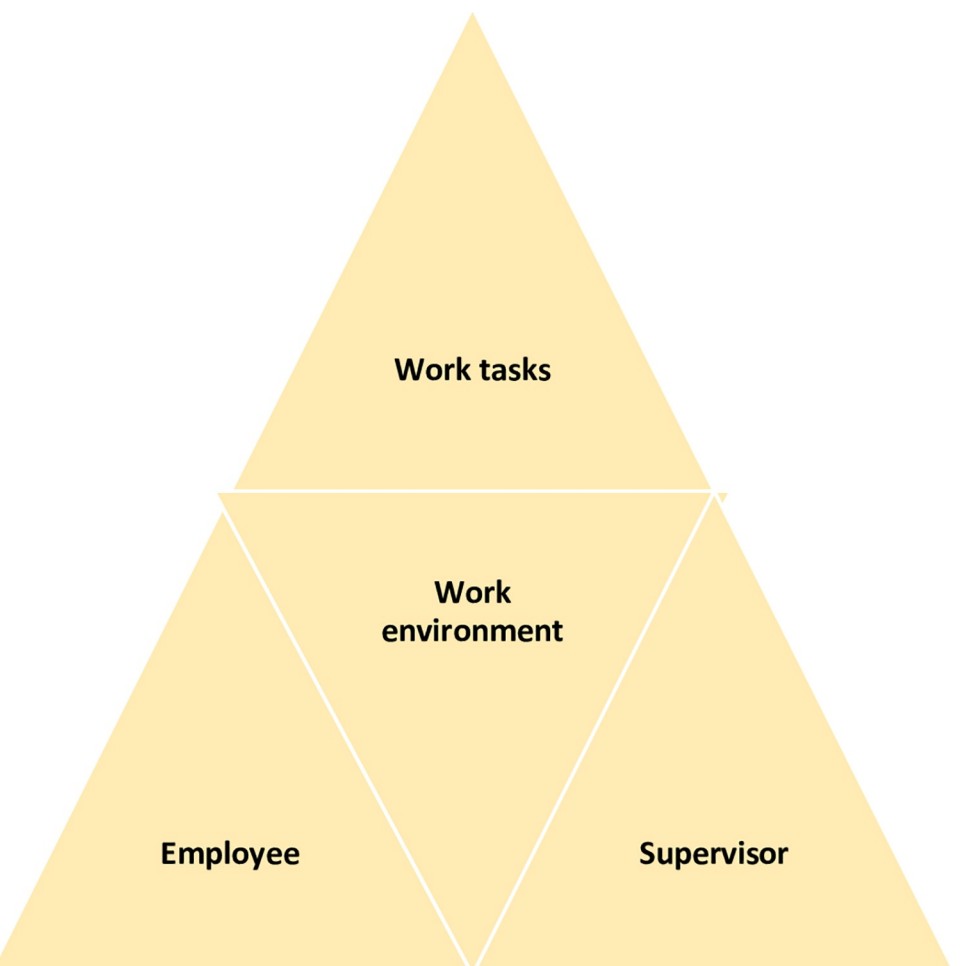

**Fig 1. The health promotion approach used in the convergence dialogue meeting.**

interviews being completed before the analysis process started, rather than the traditional emergent design.

In the first step of the analysis, two of the researchers (the first author and SK) uninvolved in data collection read the transcripts carefully and listened to the recordings repeatedly. Ideas that related to the emerging results were continuously written down in memos that were used in the analysis. Both researchers performed separate open coding of the transcripts using the Open Code software [31]. This means that the transcripts were read line by line and paragraph by paragraph. Important information was given a code with a low level of abstraction. The two researchers regularly discussed the open codes in relation to the interview transcripts, and this ensured that the emerging analysis was grounded in the empirical data. The researchers independently compiled open codes with similar content into categories on a more abstract level. Axial coding was used to indicate relationships between the categories. Codes and categories were discussed with a third researcher (AFW) who had read a small sample of the interviews. After this step, a theoretical coding process was undertaken. The aim was to build a theoretical model. Categories and memos were interpreted once more, with higher abstraction to sensitize concepts that were building the model [30]. Through the analysis process, triangulation between researchers with different backgrounds was used [31]. The research group had backgrounds in work rehabilitation, stress-related

**Table 1. Core category, phases and properties resulting from the analysis.**

| Core category/process | Restoring confidence on common ground | | |
|---|---|---|---|
| **Phases** | Emotional entrance | Supportive guidance | Empowering change |
| **Properties** | Vulnerability | Competence | Heading toward confidence |
| | Anxiety/distress | Coordination | Transferring knowledge |
| | Expectations | Balancing power | Improved collaboration |

and pain rehabilitation, insurance medicine, psychiatry, gender science, physiotherapy, occupational health, pedagogy and education.

## Results

The analysis resulted in a core category, *restoring confidence on common ground*, three sub-categories presented as phases: *emotional entrance*, *supportive guidance* and *empowering change*. Each phase included three properties or underlying components (Table 1).

The core category with its phases built a theoretical model that presents participant-empowered progress to RTW. We interpret this as a health promoting process (Fig 2) directed towards a sustainable RTW for persons with SED. This process relies on participant participation in the 24-week multimodal rehabilitation program and social and relational surroundings. Examples of such surroundings include the family, relations at the workplace, the societal view of SED, mental health issues, and sick leave.

The core category, *restoring confidence on common ground*, described participant progress from the *emotional entrance* when they started to prepare for RTW, through experience of the *empowering change* when they became safe in the RTW process due to the intervention's *supportive guidance*. The supportive guidance relies upon the structured three-step interview model and guided support from the rehabilitation coordinator. The properties of each phase describe components that have meaning for participant progress. Each phase differed in space and time, and depended on participant and workplace prerequisites and need for support. The common ground represents how supported communication and common planning balance

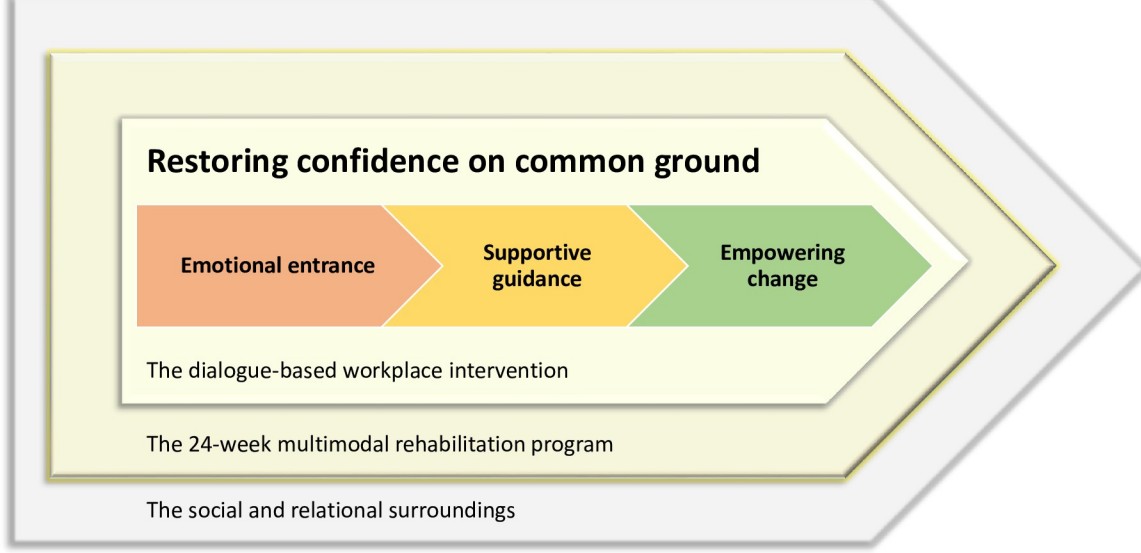

**Fig 2. Illustrated theoretical model with its health promoting process directed towards a sustainable return to work.**

relations between persons involved in the RTW process. The rehabilitation coordinator's neutral and supportive role was central in participant experiences of participation and shared responsibility in the intervention.

## Emotional entrance

The first phase, *emotional entrance*, is comprised of three properties: *vulnerability*, *anxiety/distress*, and *expectations*. This phase represents the participants' emotional experiences reflecting on their workplace as they start to prepare for RTW. We interpret this phase as visualizing the emotional exposure that people with SED experience related to the cause of their illness and the meaning of the work environment when planning for RTW.

*Vulnerability* captures participant experiences of personal failure, loss of control, and causing problems for colleagues and family. Participants were "ashamed of being exhausted" and blamed themselves for "being weak and not able to manage as much as others". They found it difficult that they "not were able to handle the situation by themselves" and needed help. Participants felt sad that they had to change behaviors they thought might have caused their illness, such as engagement and responsibility at work. Overall, being in a vulnerable and emotional situation made them doubt their ability to return to work.

*Anxiety/distress* represented participant concerns about their relation to the workplace and the supervisor. Participants worried about what the supervisor might have told the rehabilitation coordinator in the individual interview, what the supervisor perceived as the reason for their exhaustion, and that they would be misunderstood. Their anxiety increased if they had experienced earlier failed attempts to RTW, if they had changed or had a poor relationship with their supervisor. Circumstances such as these contributed to distress before the convergence dialogue meeting. One participant described these expectations as "standing at the precipice":

> I think people stress themselves unnecessarily before the meeting. Of course, you don't know, it's like you're standing at the precipice. You don't know what to expect. Then when you finally get into the meeting and you can get started, then you calm down. However, this kind of meeting is stressful because it stands for something, you have no idea of how it will be go, and how it will be received (Works with data, 50 year old).

An additional distress was if they distrusted the supervisors' capability to understand, see, handle, and acknowledge problems related to the work environment. The participant below reflected on situations of "not being heard" and times when complaints about the work environment were not taken seriously:

> I actually tried to tell my boss, while I was away, that there was no substitute for me, although I was working more than full time. The boss only said, "No, we're just going to assign that out to everyone else". But "everyone else" were those who didn't have time to help me before, because they already had enough to do. They told me it was tough, but they did not say that to the boss, of course. But I tried to tell him that "they don't feel good". But he just answered, "No, no one has said anything to me. I've never heard it, so there is no problem." (Works with things, 33 year old)

*Expectations* encompassed participant intentions to RTW. Participants were grateful for getting help, support and guidance in this process. They also told of excitement to meet with the supervisor in the convergence dialogue meeting.

## Supportive guidance

The second phase, *supportive guidance*, captured three properties: *competence, coordination, and balancing power*. This phase represented participant experiences of the role of the rehabilitation coordinator in the intervention and the convergence dialogue meeting. This phase explains the supportive qualities of personal guidance and structured support. Moreover, the rehabilitation coordinator's neutrality was central to participant experiences of the convergence dialogue meeting.

*Competence* represents participant experiences of practical and emotional support from the rehabilitation coordinator in contacts with the supervisor and Social Insurance officials. Presence of the rehabilitation coordinator was described as having someone "on my side", "backing me up" or "not being alone". Specifically, the rehabilitation coordinator brought self-confidence to the meeting with the supervisor:

> *Participant*: *You get support. When you find yourself in this situation with your employer, you feel that you have someone on your side who is a support. If you did not have that support, you would not be as sure of yourself. That's the most important part.*
>
> *Interviewer*: *And this support, is it emotional or is it that something has been expressed (by the rehabilitation coordinator) to the employer that was important to you?*
>
> *Participant*: *I think the most important thing is the emotional difference. There is certainly another support, but I think the most important thing is the emotional side, having someone to lean on. If the employer resists, then you have nothing to come up with as an employee, anywhere. If there is someone who can speak about the illness or whatever, well. . . of course it's easier. I think so. (Works with data, 50 year old).*

The rehabilitation coordinator's specific knowledge about SED gave the participants confidence that they had someone who "gave voice" to their needs and present situation. An example is acknowledging impairments related to the diagnosis.

*Coordination* was central to participant experiences of supportive guidance. The rehabilitation coordinator made participants feel safe arranging and carrying out meetings, and keeping structure and pace by "holding the reins". The rehabilitation coordinator also "made things happen" by being a deliverer, catalyst, or energizer. In the convergence dialogue meeting, the rehabilitator coordinator was helpful in prioritizing and organizing things when planning for RTW.

> *When we were all sitting there, the three of us reached an agreement. This is how it feels and. . ., it is roughly here where we should be when we start the return (to work process). In my case, I was the only person with my duties, so we started to have a little look at which ones I should pick out, what seemed suitable, what priority tasks had, and how much responsibility should there be, and. . . Well, that's how it is. You slowly get going. . . (Works with data, 49 year old).*

*Balancing power* represents experiences of the rehabilitation coordinator as a mediator who balanced relationships and questions of responsibility. Participants perceived the rehabilitation coordinator as supportive of the supervisor, which made it possible to improve poor relationships, as described below:

> *Even if you have a poor relationship, you need to come together in some way. . . It cannot get worse if you go [to the meeting] with a skilled mediator who is doing the talking and who sees*

*both sides. The rehabilitation coordinator is prepared and knows what we both think. And then we meet and go through it, [the rehabilitation coordinator] brings order to the relationship (Works with people, 38 year old).*

One participant thought the convergence dialogue meeting left him with the feeling of becoming the initiator and taking control of his situation. This was in contrast to some participants' earlier experiences of being in "second place" or "one step behind". This experience exemplifies the rehabilitation coordinator's opportunity to balance power, and increase participant experiences of participation when planning for RTW.

However, participants experienced the rehabilitation coordinator's role of balancing power as uncomfortable in situations that were demanding or required certain skills (such as being a calm and safe person, or experienced in handling difficult situations). One participant, who had previously experienced a complicated RTW process with severe conflicts with the supervisor and workplace, highlighted the rehabilitation coordinator's difficult role and the need for having the "right education and engagement", such as therapeutic conversational skills.

### Empowering change

The third phase, *empowering change*, encompassed participant experiences of the intervention, such as carrying out agreements made in the written plan, follow-ups, and starting to have expectations for the future. This phase consisted of three properties: *heading toward confidence*, *transferring knowledge*, and *improved collaboration*. This phase explains personal progress toward RTW, as well as experiences of progress among supervisors and workplace colleagues.

*Heading toward confidence* represents experiences of being trusted and encouraged "to take steps forward" and how support from the intervention helped them to successfully RTW.

*It was positive, because all the time I found it so hard to imagine that I could get back to work at all. I thought that I would have to leave. It was good that you managed to make small plans and find a solution that worked, it was great (Works with things, 33 years old).*

Participants described that following a written plan supported them in the practice of strategies they learned in multimodal rehabilitation. Examples include to "slow down", "take breaks", "breathe", and "set limits". Practicing such strategies was helpful in formulation of a realistic plan, such as finding balance in energy use during the day, prioritizing the amount of workload, and developing new routines. The written plan and follow-ups supported participants in efforts to "keep on track", and recall adjustments that were made, and by whom. Writing down agreements and solutions clarified questions of responsibility and opened up the opportunity to "talk about" problems in the workplace. In addition, the written plan and follow-ups implied regular evaluations, which made it possible for participants to notice their own progress:

*It's been great to be able, during the course of work to. . . discuss at meetings that this happened while I was working and it felt scary. And to find out why it happened and what can be done, and things like that. I think that it has made a huge difference between how you felt the first day or the first week when you worked, and how you feel today (Works with people, 37 year old).*

*Transferring knowledge* represents experiences of changed approaches and behaviors among supervisors and colleagues. Participants thought the intervention educated supervisors

about SED and related impairments, such as difficulty concentrating, memory problems or fatigue. In addition, participants experienced that supervisors increased their knowledge about work rehabilitation issues, for example making plans and agreements, or making relevant adjustments in the work environment. Increased knowledge raised supervisor awareness, and that contributed to an improved work environment and changed attitudes toward persons on sick leave. One participant describes how she had to revise her view of her supervisor:

> *Previously people thought. . .if you get sick, well, you have to look for a job somewhere else. . .. That you weren't worth anything. That was my biggest fear. . ..but it has just turned out the opposite. The talks and meetings have changed the way she [the boss] looks at those of us who are on sick leave and I've been treated incredibly well. I hadn't expected this. I've also heard others at my workplace say they have also been treated better. I think she's learned something from this too. Actually. . .I have been given backing, and she has met and spoken privately with the coordinator. The communication between us is now very different. She has followed up and checked how things are going, and what we have decided, which I don't think we would have done otherwise. (Works with people, 37 year old)*

Participants described *improved collaborations* with the supervisor, workplace, and Social Insurance Office. Experiences of clarified roles and allocated responsibility for each person involved increased participant safety in RTW. Improved collaboration empowered participants to discuss weaknesses and implement relevant adjustments at the workplace. Participants felt that the Social Insurance officials trusted the intervention, and this also made them feel secure. Overall, participants experienced that the intervention had a "ripple effect" by positively influencing stakeholders involved in the process. Having good relationships with involved stakeholders facilitated RTW.

Nevertheless, participant expectations for the future also involved worries, for example about their ability to handle future challenges. One participant expressed the wish to have "a manual" to prevent failures or setbacks such as returning to sick leave:

> *Well, if you get a setback, maybe not having to call someone, but if there could be a manual for what to do. It's hard to find a manual that covers everything but. . . something for future generations. . . that there is something you can do, and whether to talk to your supervisor or not (Work with data, 50 years old).*

To avoid failures and sustain changes, participants suggested having longer support or having an additional convergence dialogue meeting in the latter part of the intervention. They were also concerned about the handoff to primary care or occupational healthcare when they finished the multimodal rehabilitation program. One participant felt that participating in the intervention not had met her expectations because her exhaustion was related to her home situation rather than her workplace.

## Discussion

The aim of this study was to explore experiences from persons with SED who participated in a dialogue-based workplace intervention with convergence dialogue meetings and personal guidance from a rehabilitation coordinator. The study provides knowledge about the importance of support of persons with SED who are preparing for RTW. All study participants returned to their previous workplaces. This shows that suffering from SED does not mean having to change jobs to successfully RTW. Elaboration on the meaning of the supportive and health promotion outcomes is interesting from a pedagogical framework, *zone of proximal*

*development* theorized by Vygotsky [35–37], and the pedagogical concept of *scaffolding* [36, 38]. Zone of proximal development is an established pedagogy used to explain how personal guidance supports learning processes and independency in individuals [39, 40]. The pedagogy emphasizes that psychological development occurs in interactive processes between the individual and social environment through independent problem solving under supervision [39, 41]. Figuratively, the zone of proximal development symbolizes the distance between knowledge that is reached or unreached. This distance is determined by the level of supervision needed by a person who is still learning compared to a more competent peer [36, 41]. In our study, the zone of proximal development explains the relationship between the participant (the person learning) and the rehabilitation coordinator (the more competent peer), and to some extent the relationships with the rehabilitation coordinator, participant, and supervisor. The pedagogy explains that the knowledgeable relationship is unequal at the beginning of a learning process and thereafter becomes balanced as the need for support from the more competent peer decreases and the learner's independence increases [37].

*Scaffolding* represents the use of practical tools to provide cognitive support in a learning process. For example, tools might be supportive information, written agreements, goal setting or software [36, 39]. In our study, participants experienced this kind of substantial support as the structure of the intervention that stimulated progress to "keep on track" by creating constructive goals, complying with written plans, and conducting follow-ups. Guided support that promotes common communication and collaboration establishes trust in a learning process [38]. Our theoretical model, *restoring confidence on common ground*, with convergence dialogue meetings and personal guidance from a rehabilitation coordinator relies on a health-promoting pedagogy that stimulates learning processes and personal development for the participant and other involved stakeholders.

Participants with SED experienced emotional exposure when they entered the intervention and started to prepare for RTW. Experiences of exposure are complex among persons with SED and mental disorders, and may include personal views of vulnerability and self-blame [42–45], and societal views and stigma related to mental disorders, exhaustion and stress [12, 46]. In addition, the high risk of long-term sickness absence due to SED and mental disorders may lead to marginalization because of job insecurity. This was expressed in a fear of being judged by the supervisor or loss of the job [13, 14, 46].

Thus, emotional exposure involves experiences of insecurity, ambivalence, and subordination. In our study, participants expressed worries about meeting the supervisor and colleagues because they blamed themselves for becoming exhausted, and because of experiences of "not being heard" and difficulties addressing problems at the workplace. A meta-synthesis found that obstacles for RTW and hindrances to implement planned solutions at the workplace involved individual factors such as high responsibility, low self-efficacy, lack of support from the workplace, and complicated organizational structures [14]. Low workplace support, including low support from supervisors and colleagues, is associated with emotional exhaustion [13]. High levels of psychological demands, including high work pace and conflicting demands at work, increase the risk for burnout [47].

Self-efficacy concerns the ability to meet demands at work with engagement, independence and participation [48]. Self-efficacy in the RTW process is encouraged through creating concrete goals and having social support from supervisors, colleagues and other authorities [48]. Our study participants felt that personal and substantial support from the rehabilitation coordinator in conjunction with support from the supervisor encouraged them to return to work. Thus, work-oriented interventions, including a pedagogical framework with structured dialogues, solution-driven planning, and goal-setting, have the potential to encourage RTW self-efficacy among persons with SED, and thereby promote prerequisites for earlier RTW.

Personal guidance from the rehabilitation coordinator built safety and stability in the RTW process. Personal guidance, by trained RTW coordinators and convergence dialogue meetings enhances RTW for persons with SED and mental disorders [14, 21, 25, 26, 49]. In our study, the participants described the rehabilitation coordinator as having a number of supportive competencies. The rehabilitation coordinator provided practical and emotional support by being someone to "lean on", as well as being an energizer and catalyst who "makes things happen". The rehabilitation coordinator "gave voice" for the participants in situations when they had problems of "being heard", or when they needed support for diagnosis-specific needs. In addition, the rehabilitation coordinator's neutrality and ability to balance unequal power relations (for example, with the supervisor or the Social Insurance official) was highlighted. The distinct competencies of the rehabilitation coordinator illustrate the value of having persons with specific competencies intimately involved in the preparation and implementation of a RTW process. A neutral role in coordinating contacts and communication between healthcare and workplace initiates the opportunity for improved collaboration [14, 25]. A study evaluating a workplace-oriented intervention with a convergence dialogue meeting and performed by a treating psychologist discussed the need for specially trained persons to perform the convergence dialogue meeting. De Weerd et al [26] said that being competent in work-related issues and the use of structured guidelines increases the prerequisites for earlier RTW [26]. These suggestions are consistent with findings from a meta-synthesis on RTW for persons with mental disorders [14]. Andersen et al [14] discuss how RTW is enhanced by integrating the cause of the sick leave among supervisors, and formulating individualized RTW plans that include modified work tasks. These findings conform to the intervention in our study that had a competent rehabilitation coordinator perform convergence dialogue meetings and formulate an individualized plan for RTW that included agreements between the supervisor and participant.

The participants emphasized that the rehabilitation coordinator needed the skills to maintain neutrality and handle conflicts during collaboration with the supervisor. Extensive expertise and appropriate personal qualities are important when working as a rehabilitation coordinator [28]. Specific skills are important to facilitate communication (e.g., being pragmatic, fair and nonjudgmental), ability to incorporate workplace adjustments and mediation, social problem solving, and having knowledge about medical disabilities and regulations [50]. The encouraging effect of having a rehabilitation coordinator follow the RTW process is a competence that is important in including healthcare as part of a rehabilitation process [28, 50]. The presence of a rehabilitation coordinator makes it possible to implement organizational and work-related changes earlier in the RTW process; this affects RTW timeframes [46].

Our intervention encouraged communication and improved collaboration between the supervisor and involved stakeholders in a respectful way. The convergence dialogue meeting and formulation of a written plan made it possible to discuss problems at the workplace, which in turn meant that responsibility for RTW became an organizational matter. To date, interventions developed to improve rehabilitation for persons with SED have shown marginal effects, and this highlights the need to develop strategies for prevention and involve the workplace in the RTW process [6]. There is also the need to bridge the intention-implementation gaps between different systems, such as healthcare, occupational and social insurance facilities [14, 51].

A meta-synthesis that explored obstacles and opportunities to full RTW among workers on partial sick leave due mental disorders found that it was difficult to implement workplace solutions even with supervisor support [14]. This gap between intentions and implementation of strategies may delay or even contribute to relapse into sickness absence. One suggestion to overcome these gaps is to provide extensive support for social integration, education, and

communication at the workplace [14, 51]. Improving collaboration with the supervisor and workplace in RTW issues is important. In Sweden, it is proposed that the rehabilitation coordinator make this contact and bring extensive support to the supervisor [52]. A qualitative study by Gunnarsson et al [53] showed that there are demands that supervisors be competent, and this hinders them from seeking support to implement workplace adjustments. In addition, supervisors express the need for professional support to coordinate contacts in the RTW process [53]. Our findings indicate that the rehabilitation coordinator is a helpful resource through contributing enhanced communication with the supervisor and other stakeholders, and giving valuable support during implementation of workplace adjustments and coordinating contacts involved in RTW.

Support to communicate and carry out plans at the workplace helped participants "keep on track" in the RTW process. Pedagogical support based on zone of proximal development and scaffolding are often used in psychological therapies to establish mutual conversations and collaborations in family therapy [40]. Sundet [41] explains that mutuality helps families in "off course" situations to support each other in order to be able to move forward in a developing process. In our study, collaboration and communication "on common ground" conforms to a similar mutual interactive progress between the participant and the supervisor in creation of common goals and strategies to provide a sustainable RTW.

Furthermore, participants felt that transmission of knowledge changed attitudes toward SED and sick leave among supervisors and workplace colleagues. Other qualitative studies confirm that increased understanding of the reasons for sickness absence have preventive effects for RTW [14, 43, 54, 55]. Hence, inviting "respectful communication" is important in order to improve conditions that change workplace attitudes and cultures [54]. An additional aspect that improves workplace culture is to acknowledge how stress is talked about and create an increased awareness of the meaning of stress as a sign of engagement at the same time that it is a sign of weakness [54, 55].

Studies show that SED benefits from work-oriented interventions [6, 25]. Finnes et al [25] discuss how SED is more related to the work situation than depression is. This may explain why work-oriented interventions are more suitable for stress-related diagnoses. Such findings indicate the need to tailor interventions to suit specific needs related to different diagnostic groups and to develop preventive measures at the workplace.

## Strengths and weaknesses of the study

The study intervention was well-organized and performed in a clinical setting. This provided us with rich and varied interview data. The different backgrounds and work experiences of participants, together with participants' familiarity with each other in the interview groups, facilitated an open climate for discussion of positive and negative experiences. On the other hand, familiarity within the groups may have hindered openness since the participants had already settled into roles in their respective multimodal rehabilitation group. Overall, the participants had positive experiences of the intervention. We are aware of that the combination of the intervention and the multimodal rehabilitation program influences interpretation of the findings. Yet, the combination seems to be beneficial as it contributed to opportunities to practice strategies and regularly evaluate participant progress in the rehabilitation process.

Triangulation between researchers was used to increase study trustworthiness and credibility [31]. The research group represented different areas of experiences and competencies. In addition, the analysis process was performed with researchers uninvolved in the data collection or multimodal rehabilitation program. The coding process was independently performed by the first and the third authors, and in close collaboration with the second author. All

researchers were involved in the analysis process, building the theoretical model, and writing the manuscript. The results were thoroughly discussed between all authors and other groups of researchers who found the results to be credible. Therefore, we consider our results trustworthy.

## Practical implications

In Sweden, a rehabilitation coordinator is available in all Swedish regions and in primary and specialty healthcare. Therefore, an advantageous dialogue-based workplace intervention could be implemented in the healthcare system and occupational healthcare settings. A similar intervention with an early workplace dialogue for persons with musculoskeletal disorders shows improved work ability over time [56]. In the future, exploring whether a dialogue-based workplace intervention could be used to prevent SED and stress-related disorders at workplaces would be interesting.

## Conclusion

A dialogue-based workplace intervention with convergence dialogue meetings, performed by a rehabilitation coordinator with experience in health promotion, provided valuable support that enhances RTW for persons with SED. The intervention's health promoting pedagogy empowered participants to progress and feel safe in the RTW process. Continuous guidance from a rehabilitation coordinator, with a structured framework that includes a convergence dialogue meeting, joint planning and follow-ups, enhanced communication and collaboration between the employee, supervisor and other involved stakeholders. Participants said that communication and collaboration balanced relationships, conveyed knowledge, and changed attitudes about SED among supervisors and workplace colleagues. Entry of a rehabilitation coordinator who performs a dialogue-based workplace intervention has a beneficial contribution that enhances RTW for persons with SED, and bridges the gaps between healthcare, the workplace, and other organizational systems. The intervention also contributes to a positive re-orientation towards a successful RTW instead of viewing SED as an endpoint for employment or a career in the workforce. In a prolonged process, a dialogue-based workplace intervention with convergence dialogue meetings and a rehabilitation coordinator may secure a sustainable RTW for persons with SED.

## Supporting information

**S1 Appendix. Structured three-step interview model–interview guide in English.** (PDF)

**S2 Appendix. Structured three-step interview model–interview guide in Swedish.** (PDF)

## Acknowledgments

The authors thank the individuals who participated in the study for sharing their experiences, personnel at the Stress Rehabilitation Clinic for engagement in the project, and specifically the personnel who collected and transcribed the data.

## Author Contributions

**Conceptualization:** Maria Strömbäck, Anncristine Fjellman-Wiklund, Sara Keisu, Marine Sturesson, Therese Eskilsson.

**Data curation:** Maria Strömbäck, Therese Eskilsson.

**Formal analysis:** Maria Strömbäck, Anncristine Fjellman-Wiklund, Sara Keisu, Therese Eskilsson.

**Funding acquisition:** Therese Eskilsson.

**Investigation:** Maria Strömbäck, Sara Keisu, Marine Sturesson.

**Methodology:** Maria Strömbäck, Anncristine Fjellman-Wiklund, Therese Eskilsson.

**Project administration:** Maria Strömbäck.

**Resources:** Therese Eskilsson.

**Software:** Maria Strömbäck, Sara Keisu.

**Supervision:** Maria Strömbäck, Anncristine Fjellman-Wiklund, Therese Eskilsson.

**Validation:** Maria Strömbäck.

**Visualization:** Maria Strömbäck.

**Writing – original draft:** Maria Strömbäck, Anncristine Fjellman-Wiklund, Sara Keisu, Therese Eskilsson.

**Writing – review & editing:** Maria Strömbäck, Anncristine Fjellman-Wiklund, Sara Keisu, Marine Sturesson, Therese Eskilsson.

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
