## [Decision Letter · Decision Letter 0]

12 Feb 2020

PONE-D-19-14603

Restoring confidence in return to work: A qualitative study of the experiences of persons with exhaustion disorder after a dialogue-based workplace intervention

PLOS ONE

Dear PhD Strömbäck,

Thank you for submitting your manuscript to PLOS ONE. After careful consideration, we feel that it has considerable merit but does not fully meet PLOS ONE’s publication criteria as it currently stands. Therefore, we invite you to submit a revised version of the manuscript that addresses the points raised during the review process.

I ask that you please carefully address the reviewer concerns in a resubmitted manuscript. 

We would appreciate receiving your revised manuscript by Mar 28 2020 11:59PM. To enhance the reproducibility of your results, we recommend that if applicable you deposit your laboratory protocols in protocols.io, where a protocol can be assigned its own identifier (DOI) such that it can be cited independently in the future. For instructions see: http://journals.plos.org/plosone/s/submission-guidelines#loc-laboratory-protocols

We look forward to receiving your revised manuscript.

Kind regards,

Adam T. Perzynski, PhD

Academic Editor

PLOS ONE

Journal Requirements:

1. If the semi-structured interview guide developed as part of this study and it is not under a copyright more restrictive than CC-BY, then please include a copy, in both the original language and English, as Supporting Information.

Review of Plos One 19-14603.

This manuscript, “Restoring confidence in return to work: A qualitative study of the experiences of

persons with exhaustion disorder after a dialogue-based workplace intervention” is an interesting qualitative study on an important topic. While most qualitative studies in this field are observational, this study is evaluative in that it examines response to an intervention. Many thanks to the authors for the opportunity to review their work. I have outlined some minor concerns I have with the manuscript that the authors should address in a resubmission.

• Abstract and manuscript overall. The abstract could use some additional careful language editing. For example, the first sentence, “Stress-induced exhaustion disorder (SED) is a primary cause for sickness absence among mental health disorders in Sweden” is difficult to understand. Perhaps the authors mean, “Stress-induced exhaustion disorder (SED) is a primary cause for sickness absence and mental health disorders in Sweden”? or "among persons with mental health disorders in Sweden"? Although such language and grammatical errors are minor there are enough contained in the manuscript to distract from the message of the paper.

• Page 12, line 114. “in the setting of a 24-week multimodal rehabilitation program” is unclear. What else (if anything) does the 24 week program include beyond the intervention being studied in the manuscript?

• The authors should probably comment briefly about why grounded theory is more appropriate than a phenomenological approach to the analysis. Given all participants have a set of shared experiences, and a shared intervention, many researchers would have selected phenomenology as the analytic approach.

• Could the structured interview guide questions from step 2 be made available as a supplemental appendix?

• A description of training for conducting CDM does not appear to have been provided in the manuscript.

• What outcome(s) are implied that the arrow is pointing toward in Figure 2?

• I struggled with the presentation of results at times. It seems that participants are sometimes reflecting on their workplace, and sometimes reflecting on the format of the intervention. This happens throughout page 12, for example. Please attend to this problem of organization of results.

• There are only a handful of quotations used in the results section to support the grounded thematic structure. It would be useful to add some additional quotations from participants supporting the core constructs especially, if available.

• It seems that one of the novel features of the program and use of a rehabilitation coordinator is the connection between the intervention team (here the rehabilitation coordinator) as liason between the worker and the supervisor. This is clearer in the results than in the description of the intervention. Please make this component of the intervention clearer in the description of the intervention.

• I think one of the major contributions of this manuscript is somewhat understated. In many countries and workplace scenarios, burnout has a feeling of finality. Workers become hopeless and burnout or SED) is viewed as an endpoint for employment and often total exit from a career or the workforce. The intervention at hand and the perspectives shared by interview participants in the current study are an important (and positive) re-orientation toward the area of workplace stress. The authors might consider emphasizing this point in the discussion or conclusion, as their study likely has considerable relevance outside of Sweden.

Reviewers' comments:

Reviewer's Responses to Questions

**Comments to the Author**

1. Is the manuscript technically sound, and do the data support the conclusions?

Reviewer #1: Yes

2. Has the statistical analysis been performed appropriately and rigorously? 

Reviewer #1: Yes

3. Have the authors made all data underlying the findings in their manuscript fully available?

Reviewer #1: No

4. Is the manuscript presented in an intelligible fashion and written in standard English?

Reviewer #1: Yes

5. Review Comments to the Author

Reviewer #1: The authors have produced qualitative research detailing the effects of an intervention that treats Stress-induced Exhaustion Disorder, which in the United States we think of as "burnout". The intervention takes place in a dedicated clinic with a designated rehabilitation coordinator. The research contributes to the scant literature on recovering from Stress-induced Exhaustion Disorder or burnout with an outcome of returning to work, rather than changing occupations. The qualitative approach helps to elaborate the perspectives of the individuals who are being treated for the disorder.

Minor comment #1: This is stated later in the manuscript, but it would be helpful on page 5 as the rehabilitation coordinator is being introduced, to describe the setting the individual works (the clinic) and some examples of what they do outside of the detailed intervention. For example, how might they help a patient recovering from a hospitalization from a heart attack or from shoulder pain from an overuse injury.

Minor comment #2: At least twice in the manuscript, CDM is replaced with "CMD". As I had to look up the meaning of the acronym each time it came up, I wonder if the manuscript would be clearer if the authors used the full term each time instead.

6. PLOS authors have the option to publish the peer review history of their article (what does this mean?). If published, this will include your full peer review and any attached files.

Reviewer #1: Yes: David Margolius

---

## [Author Response · Author response to Decision Letter 0]

14 Apr 2020

PONE-D-19-14603

Restoring confidence in return to work: A qualitative study of the experiences of persons with exhaustion disorder after a dialogue-based workplace intervention

PLOS ONE

Response letter 

Dear Academic Editor Adam T Perzynski, 

Thank you for giving us the opportunity to submit a revised version of the manuscript “Restoring confidence in return to work: A qualitative study of the experiences of persons with exhaustion disorder after a dialogue-based workplace intervention”. 

We have read and considered the points raised during the review process and we are very grateful for the help to improve our manuscript according to your suggestions. 

In the following, we have responded yours and reviewers’ comments point by point. 

Sincerely,

On behalf of co-authors 

Maria Strömbäck, PhD, PT, communicating author 

Institution of Community Medicine and Rehabilitation, Physiotherapy, 

Umeå University, Sweden

Comments from editor:

If the semi-structured interview guide developed as part of this study and it is not under a copyright more restrictive than CC-BY, then please include a copy, in both the original language and English, as Supporting Information.

--- Authors response: We have included a copy of the structured interview guide, one in Swedish and one in English, as Supporting Information.

We note that you have indicated that data from this study are available upon request. PLOS only allows data to be available upon request if there are legal or ethical restrictions on sharing data publicly. 

If there are ethical or legal restrictions on sharing a de-identified data set, please explain them in detail (e.g., data contain potentially identifying or sensitive patient information) and who has imposed them (e.g., an ethics committee). Please also provide contact information for a data access committee, ethics committee, or other institutional body to which data requests may be sent.

--- Authors response: The Swedish Data Protection Act (1998:204) does not permit sharing de-identified and sensitive data on humans (like in our interviews). Readers may contact Maria Strömbäck to request the data. Data will be available upon request to all interested researchers.

Comments from reviewer 1

This manuscript, “Restoring confidence in return to work: A qualitative study of the experiences of persons with exhaustion disorder after a dialogue-based workplace intervention” is an interesting qualitative study on an important topic. While most qualitative studies in this field are observational, this study is evaluative in that it examines response to an intervention. Many thanks to the authors for the opportunity to review their work. I have outlined some minor concerns I have with the manuscript that the authors should address in a resubmission.

Abstract and manuscript overall. The abstract could use some additional careful language editing. For example, the first sentence, “Stress-induced exhaustion disorder (SED) is a primary cause for sickness absence among mental health disorders in Sweden” is difficult to understand. Perhaps the authors mean, “Stress-induced exhaustion disorder (SED) is a primary cause for sickness absence and mental health disorders in Sweden”? or "among persons with mental health disorders in Sweden"? Although such language and grammatical errors are minor there are enough contained in the manuscript to distract from the message of the paper.

--- Authors response: Thanks for your comment. We have made an additional language editing and looked for language and grammatical errors. In accordance to your suggestion, we have changed the sentence in the abstract (page 3, line 23), see below: 

Stress-induced exhaustion disorder (SED) is a primary cause for sickness absence among persons with mental health disorders in Sweden.

Page 5, line 126 “in the setting of a 24-week multimodal rehabilitation program” is unclear. What else (if anything) does the 24-week program include beyond the intervention being studied in the manuscript.

--- Authors response: We have removed the sentence from the introduction because the 24-week MMR program is explained later in the Method section (page 7, line 157-163), see below: 

The multimodal rehabilitation program included group-based cognitive behavioral therapy (CBT) to support behavioral changes and to develop strategies to promote health. The CBT groups consisted of eight persons in each group who met weekly for 22 three-hour sessions. Each person had two individual meetings with the group leader to set and evaluate individual goals. Examples of individual goals could include development of strategies to improve one's capability to handle stressful situations, manage high workloads, set limits or take regular breaks.

The authors should probably comment briefly, about why grounded theory is more appropriate than a phenomenological approach to the analysis. Given all participants have a set of shared experiences, and a shared intervention, many researchers would have selected phenomenology as the analytic approach.

--- Authors response: Thanks for the comment. The research group has competence in phenomenology but for this study, we made an active choice to use grounded theory. In the manuscript, we have developed the description of the theoretical base in grounded theory and added that we are inspired by Charmaz and symbolic interactionism (page 6, line 140-143). We hope that the description clarifies why we find the analytical approach appropriate when investigating social interactions, see below:

Grounded theory is theoretically based in social constructivism, and the grounded theory method used in this study is inspired by the methodology of Charmaz and symbolic interactionism [30]. Symbolic interactionism is a dynamic theoretical perspective that assumes that people construct identities, social rules and reality through interaction [30].

Could the structured interview guide questions from step 2 be made available as a supplemental appendix?

--- Authors response: Yes, the interview guide in the structured three-step interview model is now available as supporting information (page 8, line 197), see below:

The interview guide to the structured three-step interview model is attached as supporting information (S1 and S2 Appendix).

A description of training for conducting CDM does not appear to have been provided in the manuscript.

--- Authors response: We have included a description of how the rehabilitation coordinator is trained for conducting a convergence dialogue meeting in the method section (page 8, line 181-184), see below: 

The rehabilitation coordinator had specific knowledge about SED and social insurance medicine, and was educated and trained in implementing convergence dialogue meetings through a written manual and practical guidance from experienced personnel in the research group and with a background in occupational health.

What outcome(s) are implied that the arrow is pointing toward in Figure 2?

--- Authors response: We have clarified the outcome of the arrow in the heading for figure 2 (page 11, line 261), which points towards sustainable return to work, see below 

Fig 2. Illustrated theoretical model with its health promoting process directed towards sustainable return to work 

I struggled with the presentation of results at times. It seems that participants are sometimes reflecting on their workplace, and sometimes reflecting on the format of the intervention. This happens throughout page 12, for example. Please attend to this problem of organization of results.

--- Authors response: Thanks. We have tried to clarify when the participants are reflecting on the workplace and the intervention throughout the result as a whole. We hope that it is clearer now, see for example page 11, line 274-278, and below: 

The first phase, emotional entrance, is comprised of three properties: vulnerability, anxiety/distress, and expectations. This phase represents the participants’ emotional experiences reflecting on their workplace as they start to prepare for RTW. We interpret this phase as visualizing the emotional exposure that people with SED experience related to the cause of their illness and the meaning of the work environment when planning for RTW. 

There are only a handful of quotations used in the result section to support the grounded thematic structure. It would be useful to add some additional quotations from participants supporting the core constructs especially, if available.

--- Authors response: We have looked through the result and added two more quotations in the subtheme “empowering change” (page 15, line 387-389 and page 17, line 430-433). See the quotations below: 

It was positive, because all the time I found it so hard to imagine that I could get back to work at all. I thought that I would have to leave. It was good that you managed to make small plans and find a solution that worked, it was great (Works with things, 33 years old).

Well, if you get a setback, maybe not having to call someone, but if there could be a manual for what to do. It's hard to find a manual that covers everything but ... something for future generations ... that there is something you can do, and whether to talk to your supervisor or not (Work with data, 50 years old).

It seems that one of the novel features of the program and use of a rehabilitation coordinator is the connection between the intervention team (here the rehabilitation coordinator) as liaison between the worker and the supervisor. This is clearer in the results than in the description of the intervention. Please make this component of the intervention clearer in the description of the intervention.

--- Authors response: Thanks, we have added more information about the rehabilitation coordinator and we found your suggestion to use the word liaison very fruitful. We have added the sentence below in the description of the intervention (page 8, line 180): 

The rehabilitation coordinator’s role was to be a liaison between involved persons in the intervention.

I think one of the major contributions of this manuscript is somewhat understated. In many countries and workplace scenarios, burnout has a feeling of finality. Workers become hopeless and burnout (or SED) is viewed as an endpoint for employment and often total exit from a career or the workforce. The intervention at hand and the perspectives shared by interview participants in the current study are an important (and positive) re-orientation toward the area of workplace stress. The authors might consider emphasizing this point in the discussion or conclusion, as their study likely has considerable relevance outside of Sweden.

--- Authors response: Thanks for this valuable comment. We have added information according to your suggestion in the abstract and in the conclusion. We also chose to use your wording as we found it very fitting (in the conclusion, page 24, line 613-616), see below:

The intervention also contributes to a positive re-orientation towards a successful RTW instead of viewing SED as an endpoint for employment or a career in the workforce.

Comments from reviewer 2 

The authors have produced qualitative research detailing the effects of an intervention that treats Stress-induced Exhaustion Disorder, which in the United States we think of as "burnout". The intervention takes place in a dedicated clinic with a designated rehabilitation coordinator. The research contributes to the scant literature on recovering from Stress-induced Exhaustion Disorder or burnout with an outcome of returning to work, rather than changing occupations. The qualitative approach helps to elaborate the perspectives of the individuals who are being treated for the disorder.

Minor comment #1: This is stated later in the manuscript, but it would be helpful on page 5 as the rehabilitation coordinator is being introduced, to describe the setting the individual works (the clinic) and some examples of what they do outside of the detailed intervention. For example, how might they help a patient recovering from a hospitalization from a heart attack or from shoulder pain from an overuse injury 

--- Authors response: We have developed the description of the rehabilitation coordinator’s work in the introduction, which includes working with different diseases or conditions (page 5, line 111-115) see below: 

The rehabilitation coordinator’s job is to have close collaboration with medical doctors and other team members involved in the patient’s treatment and rehabilitation plan, including mapping of medical needs, current work and earlier periods of sick leave. Depending on setting, the rehabilitation coordinator has a broad range of knowledge in the natural history of different diseases and conditions (for example SED), the rehabilitation process, assessments made in healthcare and insurance medicine, the labor market, and work-related rehabilitation efforts [28].

Minor comment #2: At least twice in the manuscript, CDM is replaced with "CMD". As I had to look up the meaning of the acronym each time it came up, I wonder if the manuscript would be clearer if the authors used the full term each time instead.

--- Authors response: Thanks, we have changed to the full term throughout the manuscript.

---

## [Editor Report · Decision Letter 1]

5 Jun 2020

Restoring confidence in return to work: A qualitative study of the experiences of persons with exhaustion disorder after a dialogue-based workplace intervention

PONE-D-19-14603R1

Dear Dr. Strömbäck,

I am very glad to inform you that your manuscript has been judged scientifically suitable for publication and will be formally accepted for publication once it meets all outstanding technical requirements.I know this has been a long journey from your initial submission to the journal. Please accept my apology for the length of the process. I am working on some efforts to streamline qualitative manuscript review. The most difficult challenge for a general journal like PLOS ONE has been in building up a cohort of reviewers competent to review such qualitative studies in health services research. You can help with this challenge by encouraging your colleagues and upper-level PhD students to agree to review. We have a near unlimited need for reviewers, so feel free to send reviewers my way. 

Kind regards,

Adam T. Perzynski, PhD

Academic Editor

PLOS ONE

Additional Editor Comments (optional):

The authors have adequately responded to peer-review concerns and the manuscript is deemed suitable for publication.
---

## [Editor Report · Acceptance letter]

20 Jul 2020

PONE-D-19-14603R1 

Restoring confidence in return to work: A qualitative study of the experiences of persons with exhaustion disorder after a dialogue-based workplace intervention 

Dear Dr. Strömbäck:

I'm pleased to inform you that your manuscript has been deemed suitable for publication in PLOS ONE. Congratulations! Your manuscript is now with our production department. 

Kind regards, 

on behalf of

Dr. Adam T. Perzynski 

Academic Editor

PLOS ONE